# Arioc: high-throughput read alignment with GPU-accelerated exploration of the seed-and-extend search space

Richard Wilton[1], Tamas Budavari[2], Ben Langmead[3,6],
Sarah J. Wheelan[4,7], Steven L. Salzberg[3,5,6] and Alexander S. Szalay[1,3]

[1] Department of Physics and Astronomy, Johns Hopkins University, Baltimore, MD, USA
[2] Department of Applied Mathematics and Statistics, Johns Hopkins University, USA
[3] Department of Computer Science, Johns Hopkins University, USA
[4] Department of Oncology, Johns Hopkins University School of Medicine, USA
[5] Department of Biomedical Engineering, Johns Hopkins University, USA
[6] Center for Computational Biology, McKusick-Nathans Institute of Genetic Medicine, Johns Hopkins University, USA
[7] Center for Computational Genomics, Johns Hopkins University, USA

## ABSTRACT

When computing alignments of DNA sequences to a large genome, a key element in achieving high processing throughput is to prioritize locations in the genome where high-scoring mappings might be expected. We formulated this task as a series of list-processing operations that can be efficiently performed on graphics processing unit (GPU) hardware.We followed this approach in implementing a read aligner called Arioc that uses GPU-based parallel sort and reduction techniques to identify high-priority locations where potential alignments may be found. We then carried out a read-by-read comparison of Arioc's reported alignments with the alignments found by several leading read aligners. With simulated reads, Arioc has comparable or better accuracy than the other read aligners we tested. With human sequencing reads, Arioc demonstrates significantly greater throughput than the other aligners we evaluated across a wide range of sensitivity settings. The Arioc software is available at https://github.com/RWilton/Arioc. It is released under a BSD open-source license.

## INTRODUCTION

The cost and throughput of DNA sequencing have improved rapidly in the past several years (*Glenn, 2011*), with recent advances reducing the cost of sequencing a single human genome at 30-fold coverage to around $1,000 (*Hayden, 2014*). It is increasingly common for consortia, or even individual research groups, to generate sequencing datasets that include hundreds or thousands of human genomes. The first and usually the most time-consuming step in analyzing such datasets is read alignment, the process of determining the point of origin of each sequencing read with respect to a reference genome. The continued growth in the size of sequencing datasets creates a crucial need for efficient and scalable read alignment software.

Corresponding author
Richard Wilton,
richard.wilton@jhu.edu

To address this need, a number of attempts have been made to develop read-alignment software that exploits the parallel processing capability of general-purpose graphics processing units, or GPUs. GPUs are video display devices whose hardware and system-software architecture support their use not only for graphics applications but also for general purpose computing. They are well-suited to software implementations where computations on many thousands of data items can be carried out independently in parallel. This characteristic has inspired a number of attempts to develop high-throughput read aligners that use GPU acceleration.

Experience has shown, however, that it is not easy to build useful GPU-based read alignment software. In general, GPU hardware is perceived as being impractical from a software-engineering standpoint for the task of computing read alignments (*Kristensen, 2011*). This impression is reinforced by the common misconception that GPU hardware provides speed improvements in direct proportion to the number of concurrently-executing GPU threads, that is, that the same amount of work "ought to" run 1,000 times faster on 20,000 concurrent GPU threads than on 20 concurrent CPU threads.

In practice, software running on GPU hardware is constrained by a variety of algorithmic and software-engineering considerations. GPU programming requires software to manage single-instruction multiple-data (SIMD) threading, but efficient handling of memory (data layout, caching, data transfers between CPU and GPU memory) also requires a great deal of attention. Such threading and memory-management constraints lead to realistic GPU-based speed improvements on the order of $10\times-100\times$ (*Anderson et al., 2011*).

The salient problem in engineering a GPU-accelerated read aligner is that the most biologically relevant sequence-alignment algorithm (*Smith & Waterman, 1981*; *Gotoh, 1982*) is not only memory-intensive but also involves dynamic programming dependencies that are awkward to compute efficiently in parallel. In general, this consideration has militated against the development of parallel-threaded GPU implementations (*Khajeh-Saeed, Poole & Perot, 2010*) where multiple threads of execution cooperate to compute a single alignment. Instead, the fastest implementations of the algorithm on both CPUs and GPUs have relied on task parallelism, where each thread of execution computes an entire pairwise alignment independently of all other parallel threads (e.g., *Carriero & Gelernter, 1990*; *Manavski & Valle, 2008*; *Liu, Wirawan & Schmidt, 2013*).

There is, however, another significant barrier to the implementation of high-throughput GPU-based alignment software. In a typical pairwise sequence alignment problem, a short (100 to 250 nt) query sequence, or "read," must be aligned with a comparatively long ($10^9$ nt or longer) reference sequence. Since a brute-force search for all plausible alignments in this setting would be computationally intractable, read aligners typically construct a "search space" (a list of reference-sequence locations) within which potential alignments might be discovered. This aspect of the sequence alignment problem accounts for a significant proportion of the computational effort involved in read alignment.

**Extract and hash subsequences ("seeds")**

```
Q: AAGCCTCCATACTTGAGTCCTGAACTGATGAA
   AAGCCTCCAT      →   0xDEA5D502
    AGCCTCCATA     →   0x29DEC1F0
     GCCTCCATAC    →   0xDB840577
      CCTCCATACT   →   0x4DBA90D5
        . . .
```

**Probe hash table to find reference-sequence locations**

```
0xDEA5D502:  01:14353363, 01:15536663, 02:06335366 ...
0x29DEC1F0:  01:14353364, 06:20159342, 18:00513566
0xDB840577:  01:14353365, 01:15536665, 05:83754151 ...
0x4DBA90D5:  (none)
```

**Compute alignments ("extend") at high-priority reference-sequence locations**

```
R: CATGTGTGAAGCCGCCATACCTGAGTCATGAAC--ATGAACTAA
            |||||||||||| |||||| ||||| | |||||
Q:          AAGCCTCCATACTTGAGTCCTGAACTGATGAA
```

**Figure 1 Seed-and-extend strategy for identifying potential alignments.** Fixed-length subsequences ("seeds") are extracted from the query sequence and hashed. Each hash value (e.g., "0xDEA5D502") is used to probe a lookup table of reference locations (e.g., "01:14353363" for chromosome 1, offset 14353363) where the corresponding seed occurs. These seed locations are prioritized and full alignments between the query sequence and the reference sequence are explored in priority order.

## Seed and extend

One algorithmic approach to exploring a reference-sequence search space is known as "seed and extend" (*Lipman & Pearson, 1985*; *Altschul et al., 1990*). The seed-and-extend technique is used in a number of successful read aligners and presents no algorithmic barriers to highly-parallel implementation.

An aligner that uses seed-and-extend relies on a precomputed index or lookup table to identify locations in the reference where a subsequence ("seed") extracted from the query sequence matches the same-length subsequence in the reference (Fig. 1). The aligner then performs a sequence-alignment computation at one or more of the reference-sequence locations it has obtained from the lookup table. In effect, the partial alignment implied by the seed match at each reference location is extended to arrive at a full pairwise alignment between the query sequence and the reference sequence.

## Frequency distribution of seed locations

Most seed sequences occur rarely in the human reference genome, but a few seed sequences inevitably occur at hundreds or thousands of different locations in the reference sequence. This is not only because certain portions of the reference are internally repetitive (e.g., homopolymers or tandem repeats) but also because short sequences occasionally occur at two or more non-overlapping positions in the reference genome (e.g., because of transposon-induced duplication). This can be illustrated for the human reference genome by plotting the frequency with which 20mers (20 nt subsequences) occur (Fig. 2).

Although the mean frequency of human 20mers is only 10.7, high-frequency 20mers account for a disproportionate percentage of reference-sequence locations in a lookup

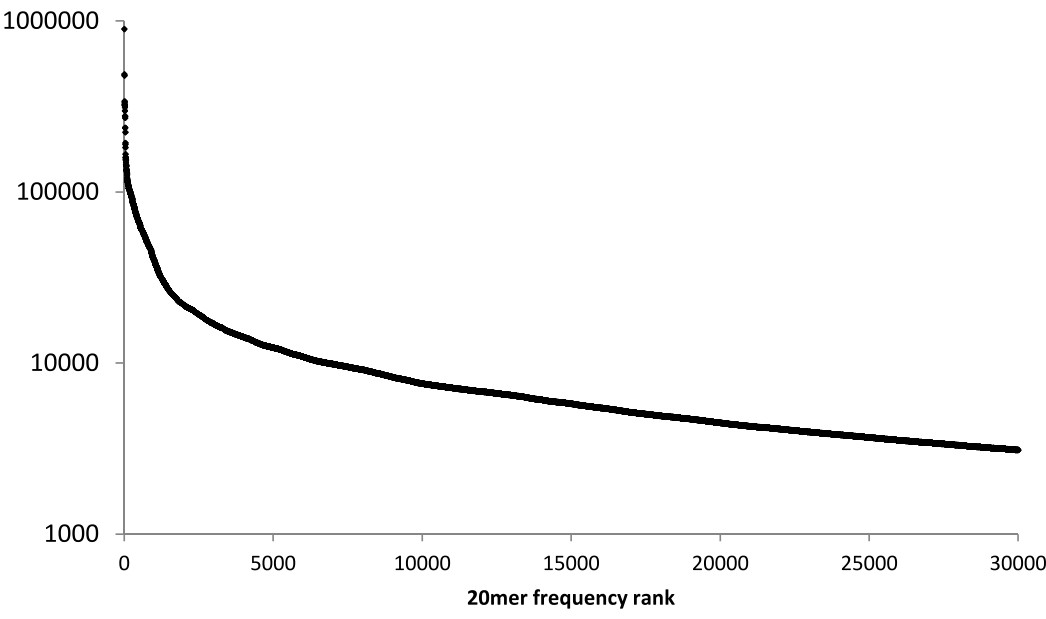

**Figure 2** **Frequency of 20mers in the human reference genome.** The number of different positions at which the 30,000 most frequently repeated 20mers occur in the human reference genome, ranked in descending order.

table. For example, only 0.1% of the 20mers in the human reference genome appear in 200 or more different locations, but they account for about 10% of the 20mers in the reference sequence. In contrast, 71.7% of the 20mers in the human genome occur exactly once.

For a read aligner that implements a seed-and-extend strategy, this long-tailed distribution of seed frequencies is a computational obstacle for reads that contain one or more "high-frequency" seeds. To avoid searching for potential alignments at an inordinate number of locations in the reference sequence, an aligner must limit the number of locations at which it computes alignments for reads that contain such seeds.

## Limiting the search space

Read aligners address this problem by using several heuristics, all of which limit the number of potential alignments computed:

- Limit the number of high-scoring mappings reported per read.
- Limit the number of seeds examined per read.
- Limit the number of reference locations examined per seed.

These heuristics trade throughput for sensitivity. An aligner spends less time computing potential alignments simply because it does not examine the entire search space (all reference locations for all seeds in each read). For the same reason, however, the aligner is less likely to identify all of the high-scoring mappings for each read. The Arioc aligner implements two different heuristics to mitigate this problem.

**Build hash table and identify "big buckets"**
```
0xDEA5D502:  01:14353363, 01:15536663, 02:06335366 ...
0x29DEC1F0:  01:14353364, 06:20159342, 18:00513566 ...
0xDB840577:  01:14353365, 01:15536665, 05:83754151 ...
0x4DBA90D5:  01:14353366, 01:01903425, 05:83754152 ...
```

**Find adjacent reference-sequence locations in adjacent "big buckets"**
```
0xDEA5D502:  01:14353363, 01:15536663, 02:06335366 ...
0x29DEC1F0:  01:14353364, 06:20159342, 18:00513566 ...
0xDB840577:  01:14353365, 01:15536665, 05:83754151 ...
0x4DBA90D5:  01:14353366, 01:01903425, 05:83754152 ...
```

**Remove adjacent locations from big buckets**
```
0xDEA5D502:  01:14353363, 01:15536663, 02:06335366 ...
0x29DEC1F0:  01:14353364, 06:20159342, 18:00513566 ...
0xDB840577:  01:14353365, 01:15536665, 05:83754151 ...
0x4DBA90D5:  01:14353366, 01:01903425, 05:83754152 ...
```

**Figure 3** **Hash-table construction: sampling in repetitive regions.** Adjacent reference-sequence locations are removed from the hash table where they are found in adjacent "big buckets" (hash-table lists whose cardinality exceeds a user-configurable threshold).

## Reference-location sampling

Arioc uses hash tables in which every location in the reference sequence is sampled. For highly repetitive regions of the human genome, however, adjacent and overlapping 20mers in the reference sequence hash to a large number of locations in the reference sequence. Repetitive regions are thus associated with long hash-table lists ("big buckets") because the 20mers corresponding to those lists refer to numerous repetitive regions elsewhere in the reference.

For this reason, the Arioc lookup tables are constructed by sampling adjacent "big bucket" hash-table lists in repetitive regions so that only one such list in 10 contains a reference-sequence location that lies within the region (Fig. 3). This sampling strategy decreases the size of large hash-table lists. The tradeoff is that a read that aligns to a particular repetitive region must be seeded in up to 10 adjacent locations to guarantee that the aligner will find a reference location within that region in a hash-table list for the read.

## Seed-coverage prioritization

At run time, Arioc implements a heuristic that prioritizes alignments where a read contains two or more seeds that map to adjacent or nearby locations in the reference. This heuristic is reminiscent of the "spanning set" method used to compute alignments in the GSNAP aligner (*Wu & Nacu, 2010*), but its implementation in Arioc is actually much simpler: within any given read, Arioc assigns higher priority to a reference-sequence locus when more seeds refer to that locus (that is, when that locus is contained in more hash-table buckets for the seeds in that read).

Arioc uses an additional heuristic in the case of paired-end reads. The aligner prioritizes potential paired-end mappings where a reference location associated with at least one seed in one of the mates in the pair lies within a user-configurable distance and orientation of at least one seed in the other mate in the pair.

Notably, these heuristics are implemented using a series of sorting and reduction operations on an aggregated list of seed locations. In a CPU-based implementation, the amount of computation required for these operations would be impractical with a reference genome the size of the human genome. In a GPU-based implementation, however, these list-based operations can be performed efficiently with a combination of cooperative parallel threading (sort, stream compaction) and task parallelism (computing seed coverage, filtering using paired-end criteria). For example, an NVidia GTX480 GPU can sort over 300 million 64-bit integer values per second.

The Arioc aligner was designed to evaluate the performance of these "GPU-friendly" heuristics. In effect, Arioc implements a pipeline in which the following operations are performed on GPU hardware for each read:

- Define the "search space" for the read; that is, compute the set of reference locations that correspond to the seeds in the read.
- Adjust the reference locations so that they correspond to the location of the seed within the read.
- Sort and unduplicate the list of reference locations.
- Count the number of seeds that reference each reference location.
- For paired-end reads, identify pairs of reference locations that lie within a predefined distance and orientation of each other.
- Coalesce adjacent seed locations so that they are covered by a minimum number of alignment computations.
- Compute alignments to identify and record high scoring mappings.

## METHODS

The Arioc aligner is written in C++ and compiled for both Windows (with Microsoft Visual C++) and Linux (with the Gnu C++ compiler). The implementation runs on a user-configurable number of concurrent CPU threads and on one or more NVidia GPUs. The implementation pipeline uses about 30 different CUDA kernels written in C++ (nongapped and gapped alignment computation, application-specific list processing) and about 100 calls to various CUDA Thrust APIs (sort, reductions, set difference, string compaction).

The development and test computers were each configured with dual 6-core Intel Xeon X5670 CPUs running at 2.93 GHz, so a total of 24 logical threads were available to applications. There was 144 GB of system RAM, of which about 96 GB was available to applications. Each computer was also configured with three NVidia Tesla series GPUs (Kepler K20c), each of which supports 5 GB of on-device "global" memory and 26,624 parallel threads. The internal expansion bus in each machine was PCIe v2.

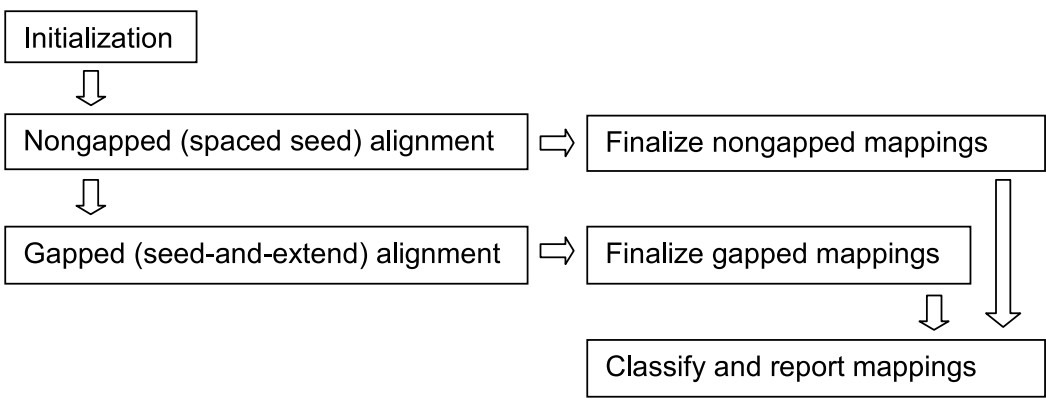

**Figure 4** **Arioc pipeline.** The Arioc pipeline implementation consists of one-time-only initialization (memory allocation, loading of lookup tables and reference data) followed by iterative batched processing of reads (query sequences). Within each batch, nongapped alignments are discovered using GPU-based spaced seed alignment. Gapped alignments, using GPU-based seed-and-extend alignment, are computed only for reads for which a satisfactory number of nongapped alignments are not found. All mappings are finalized (scored and mapped), classified, and reported in multiple concurrent CPU threads.

Throughput (query sequences aligned per second) was measured only when no other user applications were using the machines so that all CPU, memory, and I/O resources were available. For experiments with simulated data, we used Mason (*Holtgrewe, 2010*) to generate 100 nt and 250 nt paired-end reads. For experiments with Illumina data, we used 100 nt paired-end Illumina Genome Analyzer data from the YanHuang genome (*Li et al., 2009*).

## Software implementation

The Arioc implementation is a pipeline (Fig. 4) in which batches of reads are processed by a sequence of discrete software modules, each of which operates on a separate CPU thread that is allocated for the lifetime of the module and then discarded. When multiple GPUs are used, each GPU is associated with its own CPU thread. Modules are designed to execute concurrently on CPU threads and on the GPU. Data common to multiple CPU threads is shared; data common to a sequence of GPU operations resides in GPU device memory without being transferred to or from CPU memory.

The execution of the Arioc pipeline consists of iterative processing of batches of reads (query sequences), where the number of reads in a batch is constrained by the amount of available GPU memory. Within each batch iteration, the GPUs are used for list processing and for the computation of alignments, while CPU threads are used concurrently for scoring, classification, and formatting of alignment results as well as for input and output. GPU code executes concurrently with CPU code wherever possible. For example, the classification, reporting, and final output of the alignment results for a batch overlaps with the beginning of processing of the subsequent batch.

## Nongapped alignment

The nongapped aligner uses periodic spaced seeds (gapped q-grams) to identify potential mappings. The seed pattern covers 84 adjacent positions with 30 "care" positions using the

```
000     (null)
001     (null)
010     N_q
011     N_r
100     A
101     C
110     G
111     T
```

**Figure 5 Binary encoding of sequence data.** Arioc packs 21 three-bit symbols into a sequence of 64-bit values. Encoding proceeds from low-order to high-order bits. For example, the 4-nucleotide sequence AACT is encoded as binary 111101100100, or hexadecimal 0x0000000000000F64.

repeating 7-bit pattern 0010111; it is fully sensitive for nongapped alignments containing up to two mismatches when it is evaluated at seven adjacent locations in a sequence (*Chen, Souaiaia & Chen, 2009*). Both the seed value and the query sequences are encoded in 64-bit packed binary values (Fig. 5) to facilitate bitwise operations.

For each of the first seven positions in each query sequence, the result of the bitwise AND between the seed value and the query sequence is packed into a 30-bit value that is used to probe a lookup table of potential alignment locations in the reference sequence. For each such location, mappings between the query sequence and the reference sequence are identified by bitwise comparison of the entire query sequence with the corresponding reference.

Nongapped mappings with mismatches near one or both ends are examined for potential soft clipping. Arioc soft-clips a nongapped mapping whenever its alignment score is higher than it would be without soft clipping. The nongapped aligner assigns a numerical score to each mapping by applying the user-specified parameters for Smith–Waterman–Gotoh affine-gap alignment.

### Gapped alignment

Arioc only performs gapped alignment with reads for which it does not find a sufficient number of nongapped alignments. The minimum number of satisfactory nongapped alignments required for a read to be excluded from further processing is a user-configurable parameter.

The gapped aligner is a straightforward implementation of the seed-and-extend strategy. To facilitate parallel computation, multiple seed locations are examined concurrently within each read. Groups of seed locations are selected iteratively. The first group of seeds is chosen so as to cover the entire read without overlapping seeds; subsequent groups are selected so as to straddle the seed positions examined in previous groups (Fig. 6).

| iteration | seed locations in read | seeds in iteration |
|---|---|---|
| 0 | 0 20 40 60 80 | 5 |
| 1 | 10 30 50 70 | 4 |
| 2 | 5 15 25 35 45 55 65 75 | 8 |
| 3 | 2 7 12 17 22 27 32 37 42 47 52 57 62 67 72 77 | 16 |
| 4 | 1 3 6 8 11 13 16 18 21 23 26 28 31 33 36 38 41 43 46 48 51 53 56 58 61 63 66 68 71 73 76 78 | 32 |
| 5 | 4 9 14 19 24 29 34 39 44 49 54 59 64 69 74 79 | 16 |

**Figure 6 Seed locations in read sequences.** Seed locations for each of six seed iterations, for 20 nt seeds in a 100 nt read.

In each iteration, the seed subsequences are extracted from the read and hashed to 30 bits using MurmurHash3 (*Appleby, 2014*). The 30-bit hash values are used to probe a hash table of reference-sequence locations. The reference locations are prioritized and Smith–Waterman–Gotoh local alignment is computed at the highest-priority locations. Reads for which a user-configured number of satisfactory mappings have been found are excluded from subsequent iterations.

Each iteration examines seed locations that straddle the locations that were processed in previous iterations; seeds are chosen at locations that are halfway between those examined in all previous iterations. (This is similar to the behavior of Bowtie 2's -R option.) In this way the cumulative number of seeds examined doubles with each iteration, but the actual number of reference locations considered remains relatively stable. With fixed-length 20 nt seeds (20mers), six "seed iterations" are required to examine every seed location in the query sequence.

## Lookup table structures

To associate seeds with reference-sequence locations, Arioc uses two pairs of lookup tables, one pair for nongapped alignment and the other for gapped alignment. Each pair of lookup tables comprises an H table with one element for each possible hash value and a J table that contains reference-sequence locations. Each table lookup is a two-step process: a read from the H table (to obtain an offset into the J table) followed by reading a list of reference-sequence locations from the J table.

## Restricting the seed-and-extend search space

To facilitate GPU-based list operations, the Arioc implementation encodes reference locations as 64-bit bitmapped values that can be represented in one-dimensional arrays. These arrays are maintained exclusively in GPU device memory where multiple CUDA kernels can access them. CUDA kernels are used to reorganize and triage reference-location lists:

- Prioritize reference locations that lie within paired-end distance and orientation constraints.
- Prioritize reference locations where overlapping and adjacent seeds cover the largest number of adjacent positions in the reference sequence.
- Exclude reference locations that have been examined in previous seed iterations.
- Identify reference locations for which acceptable mappings exist and for which criteria for paired-end mapping are met.

## Mapping quality (MAPQ)

For each mapped read, Arioc computes an estimate of the probability that that read is mapped to a reference location other than the location where the read actually originated. MAPQ is reported as $-10\log_{10}(p)$, where $p$ is the aligner's estimate of the probability that the read is not mapped to the correct reference location. Arioc estimates $p$ using a computational model based on a probabilistic analysis of different types of mapping errors (*Li, Ruan & Durbin, 2008*). Arioc supports two user-selectable implementations of this model: one based on the methodology used in BWA-MEM (*Li, 2013*) with MAPQ scores in the numerical range 0 to 60, and another derived from the empirical logic used in Bowtie 2 (*Langmead & Salzberg, 2012*), with reported MAPQ values between 0 and 44.

## Specific concerns for GPU implementation

Available memory and computational resources on GPU devices constrain the implementation of the Arioc pipeline. Although the compiled code is not "tuned" to a particular GPU device, the source-code implementation follows programming practices that experience has shown lead to higher performance: judicious use of GPU memory and use of data-parallel algorithms and implementation methods.

### *Memory size*

The limited amount of on-device GPU memory constrains the amount of data that can be processed at any given time on a GPU. Because GPU memory requirements vary as data moves through the implementation pipeline, it is impossible to provide for full usage of available GPU memory at every processing step.

The approach taken in Arioc is to let the user specify a batch size that indicates the maximum number of reads that can be processed concurrently. In computations where available GPU memory is exceeded (for example, in performing gapped local alignment), Arioc breaks the batch into smaller sub-batches and processes the sub-batches iteratively.

Arioc also uses about 65 GB of page-locked, GPU-addressable host-system memory for its lookup tables. Data transfers from this memory are slow because they move across the PCIe bus, but the data-transfer rate is acceptable because comparatively little data is transferred during hash-table lookups.

### *Memory layout*

The Arioc implementation pays particular attention to the layout of data in GPU memory. Memory reads and writes are "coalesced" so that data elements accessed by adjacent

groups of GPU threads are laid out in adjacent locations in memory. Arioc therefore uses one-dimensional arrays of data to store the data elements accessed by multiple GPU threads. Although this style of in-memory data storage leads to somewhat opaque-looking code, the improvement in the speed of GPU code is noticeable (sometimes by a factor of two or more).

### Minimal data transfers between CPU and GPU memory

Although data can theoretically move between CPU and GPU memory at speeds determined by the PCIe bus, experience has shown that application throughput is decreased when large amounts of data are moved to and from the GPU. For this reason, Arioc maintains as much data as possible in GPU memory. Data is transferred to the CPU only when all GPU-based processing is complete.

### Divergent flow of control in parallel threads

Divergent flow of control in adjacent GPU threads can result in slower code execution. Branching logic is therefore kept to a minimum in GPU code in Arioc. Although this problem was encountered in previous GPU sequence-aligner implementations (*Schatz et al., 2007*), it is empirically less important in the Arioc implementation than the effect of optimized GPU memory access.

## Analysis of alignment results

We used the human reference genome release 37 (*Genome Reference Consortium, 2014*) for throughput and sensitivity experiments. We evaluated published results for a number of CPU-based and GPU-based read aligners (Supplementary Table T1) and identified four whose speed or sensitivity made them candidates for direct comparison with the Arioc implementation. These included two widely-used CPU-based read aligners and two recent GPU-based implementations (software versions listed in Supplementary Table T1):

- Bowtie 2 (*Langmead & Salzberg, 2012*) (CPU)
- BWA-MEM (*Li, 2013*) (CPU)
- SOAP3-dp (*Luo et al., 2013*) (GPU)
- NVBIO (*NVidia, 2014*) (GPU)

We parsed the SAM-formatted output (*SAM/BAM Format Specification Working Group, 2013*) from each aligner and aggregated the results reported by each aligner for each read. We examined the POS (position), TLEN (paired-end fragment length), and AS (alignment score) fields to ensure the consistency of the set of mappings reported by each aligner. For SOAP3-DP, which does not report alignment scores, we derived scores from the mapping information reported in the CIGAR and MD fields. We computed local alignments using the following scoring parameters: match $= +2$; mismatch $= -6$; gap open $= -5$; gap space $= -3$, with a threshold alignment score of 100 (for 100 nt reads) or 400 (for 250 nt reads).

We used simulated (Mason) reads to evaluate sensitivity for both paired-end and unpaired reads. For each aligner, we used high "effort" parameters so as to maximally

favor sensitivity over throughput. For each read mapped by each aligner, we compared the POS and CIGAR information reported by the aligner with the POS and CIGAR generated by Mason. We assumed that a read was correctly mapped when, after accounting for soft clipping, one or both of its ends mapped within 3 nt of the mapping generated by Mason. (Supplementary Table T4 explains our choice of a 3 nt threshold.) To illustrate sensitivity and specificity, we plotted the cumulative number of correctly-mapped and incorrectly-mapped reads reported by each aligner, stratified by the MAPQ score (*Li, Ruan & Durbin, 2008*) for each read.

We used the YanHuang data to measure throughput using both paired-end and unpaired reads. For this analysis, we recorded throughput across a range of "effort" parameters chosen so as to trade speed for sensitivity. We defined "sensitivity" as the percentage of reads reported as mapped by each aligner with alignment score (and, for paired-end reads, TLEN) within configured limits.

Prior to computing alignments, all of the GPU-aware aligners spend a brief period of execution time initializing static data structures in GPU device memory. We excluded this startup time from throughput calculations for these aligners.

## RESULTS

Each of the read aligners we tested is able to map tens of millions of reads to the human genome in an acceptably short period of time. All of the aligners, including Arioc, were capable of mapping simulated reads with high accuracy. With sequencer reads, Arioc demonstrated up to 10 times higher throughput across a wide range of sensitivity settings.

### Evaluation with simulated data

With simulated Illumina read data, Arioc mapped paired-end reads to their correct origin in the reference genome with sensitivity and specificity comparable to all four of the aligners to which we compared it (Fig. 7 and Supplementary Figures S1–S8). Although each aligner uses a slightly different computational model for MAPQ, all of the aligners maintain a very high ratio of correct to incorrect mappings until mappings with relatively low MAPQ scores are considered.

### Evaluation with sequencer-generated data

We used experimental data from the YanHuang human genome project to evaluate speed (Fig. 8 and Supplementary Figure S9). Across a wide range of sensitivity settings, Arioc's speed on a single GPU is about 10 times that of the CPU-based aligners to which we compared it, and two to three times that of the GPU-based aligners to which we compared it.

With this data, throughput decreases with increasing sensitivity for all of the aligners, with a steep decrease near each aligner's maximum sensitivity. This is apparent even with BWA-MEM and SOAP3-dp, although we were unable to "tune" these aligners across as wide a range of sensitivity settings as the others.

When executed concurrently on multiple GPUs in a single machine, Arioc's throughput increases in proportion to the number of GPUs (Fig. 9). At lower sensitivity settings, overall throughput is limited by PCIe bus bandwidth. Scaling improves at higher

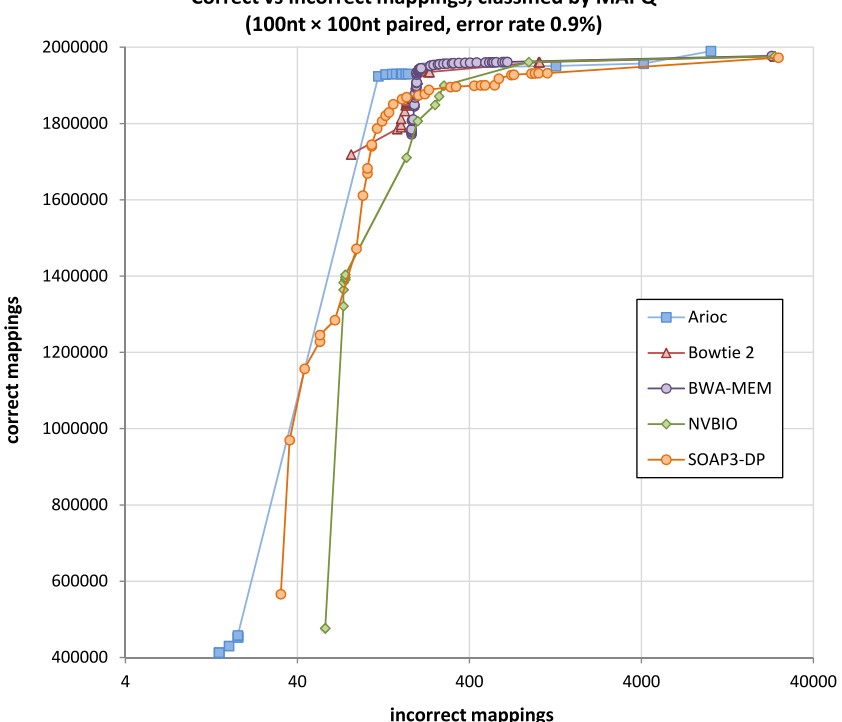

**Figure 7** **Correctly-mapped versus incorrectly-mapped simulated reads.** Total correctly mapped versus incorrectly mapped reads, plotted for decreasing MAPQ, for 1 million simulated 100 nt paired-end Illumina reads (2 million total reads). Results for unpaired reads and for 250 nt reads are similar (Supplementary Figures S1–S8).

sensitivity settings, where throughput is limited by the number of dynamic-programming computations carried out on the GPUs.

## DISCUSSION

Apart from its potential for high throughput, the Arioc implementation demonstrates that an increase in throughput can be achieved without losing sensitivity. Furthermore, by sacrificing throughput, Arioc can be "pushed" to a comparatively high level of sensitivity.

### Performance characteristics

The shape of the speed-versus-sensitivity curves we observed illustrates that read aligners achieve increased sensitivity by exploring a proportionally larger search space per successful mapping. Arioc's search-space heuristics cause it to find high-scoring mappings (that is, perfect or near-perfect alignments) rapidly within a relatively small search space. For reads that do not map with high alignment scores, however, Arioc must explore more seed locations and compute more dynamic programming problems before it can report a satisfactory mapping. For example, in the experiment shown in Fig. 8, Arioc computed about 8 times as many dynamic-programming alignments at the high end of its sensitivity range as it did at the low end of the range.

**Peer**J

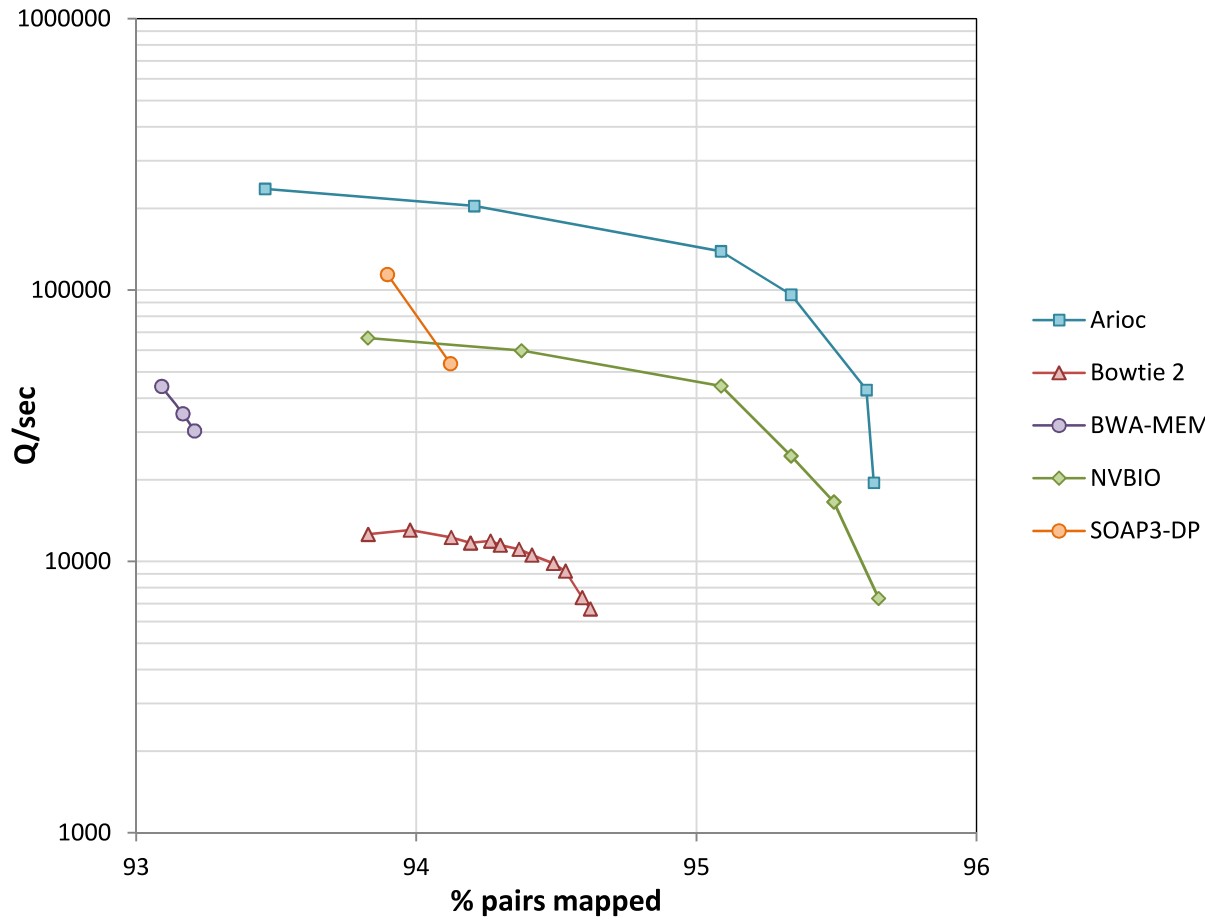

**Figure 8 Throughput versus sensitivity.** Speed (measured as the number of 100 nt query sequences processed per second) plotted versus sensitivity (expressed as the overall percentage of mapped pairs). Data for 10 million 100 nt paired-end reads from the YanHuang genome. Workstation hardware: 12 CPU cores (24 threads of execution), one NVidia K20c GPU. Results for unpaired reads are similar (Supplementary Figure S9).

Arioc explores a significantly larger search space for reads that it cannot align with a comparatively small number of mismatches or gaps. This mitigates the effect of the heuristics that filter the list of potential mapping locations on the reference sequence. In particular, gapped mappings that might be missed in an early seed iteration, when seeds are spaced widely, are detected in later seed iterations. The nature of these heuristics, however, implies that the additional mappings that Arioc finds when it is configured for higher sensitivity are generally lower-scoring mappings.

The effect of Arioc's heuristics on the computation of MAPQ (mapping quality) for a read is difficult to determine. In some cases, Arioc assigns a lower MAPQ (higher probability that the read is incorrectly mapped) simply because it computes alignments in parallel for the read and therefore tends to find more alternative mappings than would a non-parallelized implementation. On the other hand, by excluding many potential reference locations (and thus potential alternative mappings) from its search space, Arioc might incorrectly assign a high MAPQ to a read. In any event, we do not observe any notable difference overall in Arioc's MAPQ scoring when compared with other aligners.

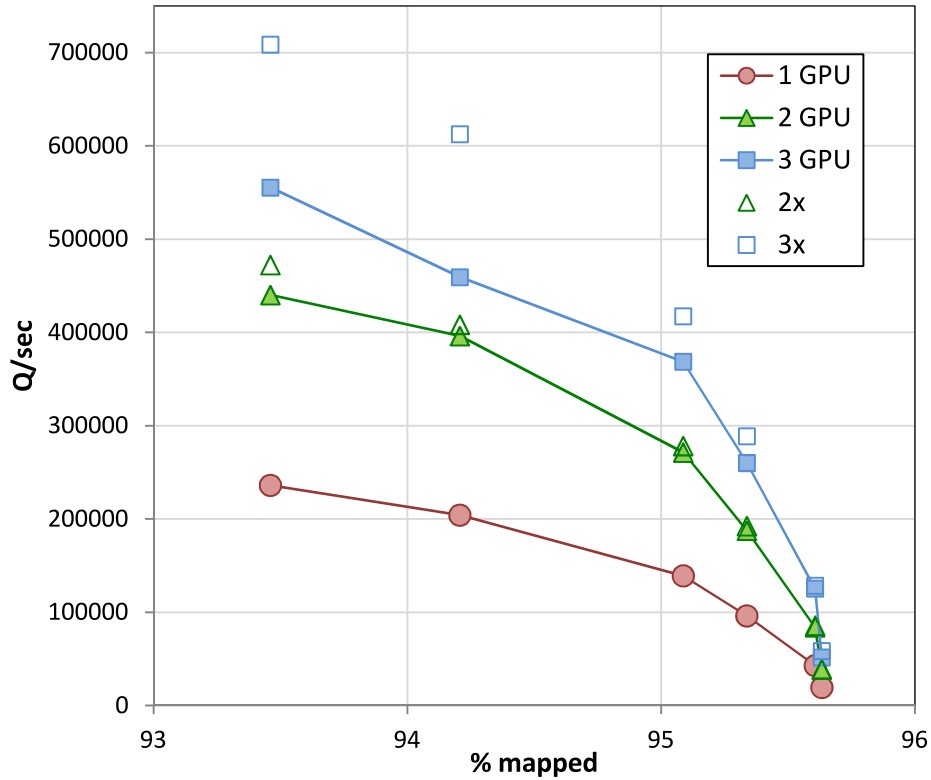

**Figure 9 Throughput on multiple GPUs.** Throughput on one, two, and three GPUs (NVidia K20c) in a single computer for the data shown in Fig. 8. For comparison, 2× and 3× multiples of single-GPU throughput are also plotted.

## GPU-accelerated sequence alignment

Unlike the Smith–Waterman–Gotoh alignment algorithm, parallel list-management algorithms—in particular, variations of radix sort and of prefix (scan) operations—are amenable to cooperative parallel-threaded implementation. Much of the intermediate processing in the sequence-alignment pipeline is thus well-suited to GPU-based implementation. We exploited this characteristic of the read-alignment process in designing and developing the Arioc software.

There are two features in the design and implementation of Arioc that distinguish it from other GPU-based read aligners. First, Arioc embodies a software design that uses the GPU as an "accelerator" in a task-parallel pipeline. CPU threads execute concurrently with GPU threads on independent data wherever possible, with synchronization points only where the GPU has completed processing a set of data. In practice, this means that overall throughput is GPU-bound and thus insensitive to variations in the time spent executing CPU threads (including post-processing alignments, reading and writing data files, and recording performance data).

Second, Arioc is the result of a software-engineering approach that emphasizes the reuse of existing code as well as the use of data structures that conform to optimal GPU memory and threading models. In particular, we emphasized data structures that can be represented

in one-dimensional arrays of integers as well as list manipulations that involve simple, data-independent numerical operations. In this way we were able to make extensive use of the NVidia Thrust library, a freely-available, well-optimized library implementation of basic parallel operations on GPU hardware. Arioc's speed derives from its use of the GPU for the kinds of computations for which the cooperative threading model is well-suited (stream compaction, radix sort, parallel prefix reduction, set difference).

We recognize that direct comparisons in speed between CPU-based and GPU-based software implementations are fraught with difficulties. We attempted to choose comparison hardware that was reasonably similar in terms of cost and availability. As more capable CPU and GPU hardware becomes available, we expect Arioc, like all of the aligners we evaluated, to deliver higher throughput.

We also foresee further optimization of Arioc's implementation. For example, there are newer, faster GPU function libraries that might be used to replace calls to the Thrust APIs. Also, we have not experimented with low-level optimization of our Smith–Waterman–Gotoh GPU implementation (*Liu, Wirawan & Schmidt, 2013*). It is likely that such optimizations will appreciably improve Arioc's throughput.

In an effort to keep up with the increasing amount of sequence data used in clinical and research settings, the usual approach to designing read alignment software has been to focus on increasing throughput. Experience with both CPU-based and GPU-based aligner implementations suggests that the most expeditious way to improve throughput is to add additional computational hardware, that is, to compute read alignments concurrently in multiple threads of execution. In this regard, therefore, GPU hardware is an attractive platform for high-throughput sequence-alignment implementations.

Nonetheless, the highly data-parallel nature of GPU hardware makes it difficult to reuse CPU-based techniques in a GPU implementation. A different approach to exploiting GPU parallelism is to use it for computational tasks that are particularly well-suited to the hardware, that are difficult to perform efficiently on sequential CPU threads, and that can improve throughput while maintaining high accuracy. Our results with Arioc demonstrate the validity of this strategy.

## ACKNOWLEDGEMENTS

We are grateful to David Luebke and Cliff Wooley of NVidia Corporation for their help in understanding some of the nuances of NVidia GPU programming.

### Funding

This work was supported by: NIH grants R01-HG007196 and R01-HG006102 (SLS); NSF grant IIS 1349906 (BL); NSF grants ACI 1261715 and ACI 1040114 (AS, RW); Gordon and Betty Moore Foundation grant 109285 (AS, RW); and JHU Discovery grants (AS, BL, SW, RW). Johns Hopkins University is an NVidia "CUDA Center of Excellence." The funders had no role in study design, data collection and analysis, decision to publish, or preparation of the manuscript.

## Competing Interests

The authors declare there are no competing interests.

## Author Contributions

- Richard Wilton conceived and designed the experiments, performed the experiments, analyzed the data, wrote the paper, prepared figures and/or tables, and reviewed drafts of the paper.
- Tamas Budavari reviewed drafts of the paper and collaborated in the design of the mapping-quality component of the software.
- Ben Langmead wrote the paper, reviewed drafts of the paper and collaborated in the design of the mapping-quality component of the software.
- Sarah J. Wheelan reviewed drafts of the paper and collaborated in the design of the dynamic-programming component of the software.
- Steven L. Salzberg wrote the paper and reviewed drafts of the paper.
- Alexander S. Szalay prepared figures and/or tables and reviewed drafts of the paper.

## Supplemental Information

Supplemental information for this article can be found online at http://dx.doi.org/10.7717/peerj.808#supplemental-information.

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
