# Peer review of "Arioc: high-throughput read alignment with GPU-accelerated exploration of the seed-and-extend search space"

_PeerJ, doi:10.7717/peerj.808_

## Round 0.1 · original submission · Major Revisions

Please consider each of both reviewers' comments carefully: The method should be described in a sufficiently formal and unambiguous way such that it is reproducible, and ideally the source code is made available. In any case, please discuss the license.

·

Basic reporting

The authors present the new GPU based read mapper Arioc. Arioc uses a hashing based seed and extend approach to prune the search space for putative optimal alignments between the reads and the reference.
Although not explicitly mentioned in the article, Arioc seems to be a best-mapper, i.e., it aims to report the best alignment (and maybe some alternatives) for each read, rather than guaranteeing to find all alignments down to a given score.
In line with this, the search space is pruned in a hierarchical way. First, ungapped matches are calculated on the GPU using lookup tables. For reads with sufficient results, alignments are computed and reported.
The other reads are investigated further, by calculating gapped matches with the reference. This is done by iteratively increasing the number of investigated seeds (here 20 mers) per read. Once an iteration provides satisfiable results, the read is excluded from further iterations.
Above strategy is implemented entirely on the GPU, and the authors motivate sufficiently why it suits the architecture.

The presented approach appears to be sophisticated and reasonably new.
I understand that this paper does not have the ambition to provide a formal description, but still the description should be more detailed sometimes.
For example, the authors talk about a seed pattern that contains 30 care positions, is stored in 64 bits and later hashed into 30 bits for the lookup.
Here, the description is quite prosaic and vague. At least, it should be defined how the nucleotides are encoded, how the care position distribution was chosen and which hash function was used.
In literature, summarizing such a seed into a limited number of care positions is also called a gapped q-gram. There exists some research about gapped q-grams, and the authors could classify their approach within this area and outline differences or similarities.

Another example for missing detail in the manuscript is the definition of the MAPQ. While the original definition of MAPQ is clear, no read mapper really applies it as it would involve to calculate all alignments of a read. However, the used simplifications are omitted often, which is also the case in this work.
Since MAPQ is a central criterion in the evaluation, the authors should describe how it is calculated with Arioc.

Experimental design

No comments.

Validity of the findings

The evaluation nicely compares Arioc to competitors without relying solely on the default parameters.
For the user of Arioc, it would be nice to see additionally what he can expect in case of using the default parameters (I suggest to additionally mark the data points for default parameters of each mapper).
While the evaluations are quite comprehensive, the presentation of the sensitivity and specificity figures (F5 and S1-S8) appears biased: in the figure presented in the main text (F5), Arioc is clearly the best. The supplementary figures for other datasets are, at least, not that obvious. While the authors do not claim to beat the competitors in this area, the choice appears a bit fishy to me.
I would suggest to put all the plots into the main paper, since, as far as I know, there is no page limit in PeerJ.

Apart form that, limiting the evaluation to four other mappers, while at least mentioning the other candidates in the supplement appears to be a reasonable choice: the purpose of this work is to present a new algorithm and provide a feeling for its performance, and not to review the entire field of read mapping algorithms.

·

Basic reporting

This is a well written manuscript about a read mapper called Arioc that efficiently uses CPUs and GPUs together. Arioc compares favorably to state-of-the-art tools. At comparable or better accuracy it seems to be 2-3X faster than other GPU-based aligners and 10X faster than CPU-only aligners.

The major problem is that the software does not seem to be open source. The PeerJ policies states that "software should be open source". I could not obtain the source code, only executable binaries. Reviewers and users are therefore unable to study the methods used in detail.

According to the license.txt file found in the Arioc.Linux.zip file, the software is distributed under the Creative Commons Attribution-NonCommercial-NoDerivatives 4.0 International Public License. I think this license is problematic in particular because derivatives are not allowed. I urge the authors to reconsider and distribute the source code under a GPL or MIT-like license.

The availability of the source code and the chosen license should be clearly stated in the manuscript.

I think the description of the seeds used by the ungapped and the gapped aligner is a bit unclear and could be improved. Perhaps an illustration of the 7 overlapping spaced seeds with their care/don’t care positions would help. A few more details about how the matches to the seeds are stored in the hash table would also improve the manuscript. Regarding the continuous seeds used for gapped alignment, an illustration of their location in the query would also help.

It is difficult to separate the dots and lines for the different tools in supplementary figures S1-S8 and S11. The use of a logarithmic scale on the x axis seemed to work out well in the similar figure 5, perhaps the same could be applied to figures S1-S8 and S11?

In figure S11 were the SNAP and YARA tools are included, an error rate of 4.3% is used. Please explain why such a high error rate was used in this case, and not 0.9% or 1.4% as in figure S5 and S6.

The error rates used in figures S1-S8 and S11 should be indicated more clearly, for instance in the figure title.

On line 297 in the manuscript I think that “Supplementary Table T2” should be “Supplementary Table T4”.

In the legend of figure S10 I assume that the reference should be to figure 6, not figure 5.

Experimental design

For the performance evaluation and comparison of tools, I think the amount of memory required or used by each tool should be indicated.

Due to the lack of source code, it might be hard to reproduce the methods.

Validity of the findings

No Comments

Additional comments

It would have been interesting to see a plot of the mapping qualities (MAPQ) computed by different aligners (e.g. Arioc vs BWA-MEM) on the same reads, correctly and incorrectly mapped.

---

## Round 0.2 · Minor Revisions

While the reviewers and I agree that the manuscript is in principle ready for publication, it remains to provide an accurate description of the notion of mapping quality used in this work, and to clarify the software and licensing issues!

·

Basic reporting

In line with the review comments, the description of the algorithm has been improved and important details have been added.

However, it remains a bit vague in case of the MAPQ:
What does "similar to BWA-MEM" mean? Is it exactly the same? If not, what is the difference? The referenced paper does not define the MAPQ but refers to the MAQ paper. Apart from that, it does not describe BWA-MEM at all. AFAIK, BWA-MEM is described in Heng Li's Arxiv paper from 2013, which does not define MAPQ either. In the same paragraph, Bowtie is mentioned but no reference is provided at all.
I would suggest to fix this section before publishing.

Apart from that, all my comments have been addressed.

Experimental design

No comments

Validity of the findings

No comments

·

Basic reporting

I think the authors have responded well to most of the issues raised. The description of the seeds is clearer now. The figures have also improved.

However, there are still a few issues which they have not yet responded well to.

1) The authors wrote in their rebuttal that the source code should be available on GitHub by 23 January. Today, a week later, the source code is still not available.

2) The abstract only states that "The Arioc software is available at: https://github.com/RWilton/Arioc";. It does not clearly state that the source code is available.

3) The manuscript does not state the type of license that applies to the software. It is still only found in a file on GitHub. It should be stated in the manuscript.

4) The authors have not commented on the choice of the "Creative Commons Attribution-NonCommercial-NoDerivatives 4.0 International Public License". As stated previously, I think this is not a wise choice of license. It is not what is considered an open source license. Also, the CC licenses are not generally intended for software. Please see the CC FAQ below for the reasons:

https://wiki.creativecommons.org/Frequently_Asked_Questions#Can_I_apply_a_Creative_Commons_license_to_software.3F

With the current state of the manuscript and the availability of the software, the software cannot be called "open source", which it should be according to PeerJ policies.

Experimental design

No Comments

Validity of the findings

No Comments

---

## Round 0.3 · accepted · Accept

While there are some problems to compile from source (pthreads), which the authors might want to solve, all questions have been answered.